# Development of a Novel LC-MS/MS Multi-Method for the Determination of Regulated and Emerging Food Contaminants Including Tenuazonic Acid, a Chromatographically Challenging *Alternaria* Toxin

**DOI:** 10.3390/molecules28031468

**Published:** 2023-02-02

**Authors:** Ádám Tölgyesi, Attila Cseh, Andrea Simon, Virender K. Sharma

**Affiliations:** 1Mertcontrol Hungary Ltd., 2144 Kerepes, Hungary; 2Program for the Environment and Sustainability, Department of Environmental and Occupational Health, School of Public Health, Texas A&M University, 1266 TAMU, College Station, TX 77843, USA

**Keywords:** pesticides, toxins, cereals, LC-MS/MS, screening, validation

## Abstract

The regulation of food contaminants in the European Union (EU) is comprehensive, and there are several compounds in the register or being added to the recommendation list. Recently, European standard methods for analysis have also been issued. The quick analysis of different groups of analytes in one sample requires a number of methods and the simultaneous use of various instruments. The aim of the present study was to develop a method that could analyze several groups of food contaminants: in this case, 266 pesticides, 12 mycotoxins, 14 alkaloid toxins, and 3 *Alternaria* toxins. The main advantage of the herein described approach over other methods is the simultaneous analysis of tenuazonic acid (TEA) and other relevant food contaminants. The developed method unites the newly published standard methods such as EN 15662:2018, EN 17194:2019, EN 17256:2019, EN 17425:2021, EN 17521:2021, which describes the analysis of both regulated and emerging contaminants. The developed method is based on a QuEChERS sample preparation, followed by LC-MS/MS analysis under alkaline mobile phase conditions. The pH of the aqueous eluent was set to 8.3, which resulted in baseline separation among ergot alkaloids and their corresponding epimers, a symmetric chromatographic peak shape for analyzing TEA and fit-for-purpose sensitivity for MS/MS detection in both positive and negative ionization modes. Those compounds, which possess the corresponding isotopically labeled internal standards (ISTD), allowed for direct quantification by the developed method and no further confirmation was necessary. This was proven by satisfactory analyses of a number of quality control (QC), proficiency test (PT), and validation samples.

## 1. Introduction

Contaminants in this study are substances which are either intentionally used in agriculture (e.g., pesticides) or which result from environmental contamination (e.g., plant toxins). Contamination may also occur during packaging, transport, or holding of foodstuffs, which causes a negative impact on the quality of food, thus risking human health. Therefore, the European Union (EU) has established maximum levels for several contaminants [1,2,3,4,5,6,7,8]. The pesticides are well-known groups of food contaminants. In the EU, the European Food Safety Authority (EFSA) assesses the safety of consumers based on the toxicity of pesticides and proposes a maximum residue limit (MRL) for their presence in food [7]. MRLs have been applied to more than 300 fresh products and to the same products after processing. Currently, the legislation covers more than 1000 pesticides recently or formerly used in agriculture worldwide. The MRL concentrations for pesticides set by the EU are summarized in regulation EC 396/2005 [6].

Mycotoxins have been regulated in the EU since 2006, beginning with well-known compounds such as aflatoxins, ochratoxin A or deoxynivalenol (DON) [2]. Subsequently, other compounds such as T-2 and HT-2 mycotoxins or citrinin have come into focus, and EU recommendations for these toxins are in force now [4]. In addition to these toxins, some other compounds known to be toxic have appeared in the EU regulations and recommendations. These so-called emerging toxins are the ergot and tropane alkaloids and the *Alternaria* toxins [1,3,5]. Our laboratory has been accredited for the standards listed above, and our aim in the present study was to combine all current methods into one novel multi-method. Hence, the simultaneous analysis of these groups of food contaminants is the focus of our current paper.

In the 1990s, the number of food contaminants analyzed simultaneously by HPLC was restricted due to the optical (HPLC-UV/FLD) or single-stage mass spectrometric (LC-MS) detection. With the widespread use of tandem mass spectrometric (MS/MS) detection in the early 2000s, a broader range of compounds could be separated in a single run, and multi-methods in food analysis have become popular [9,10]. One well-known LC-MS/MS multi-toxin method was published by Sulyok et al. in 2006 [11]. This method described the determination of 39 components employing a simple dilute-and-shoot approach. The extraction solvent for multi-mycotoxin analysis (acetonitrile-water-acetic/formic acid, 79/20/1, *v*/*v*/*v*) recommended by Sulyok has become a general extraction medium in control laboratories, and the recently published standard method (EN 17194:2019) included this solvent composition [12].

In addition to mycotoxins, the single pesticide group-based (e.g., chlorinated or phosphorated pesticide) methods have also been modified to multi-methods using both LC-MS/MS and GC-MS/MS techniques [13,14,15]. This required a general sample manipulation that can be used for all pesticides. The QuEChERS (Quick, Easy, Cheap, Efficient, Rugged, and Safe) sample preparation, which utilized acetonitrile-based extraction and phase partition, allowed for the extraction of a number of medium-polar or non-polar molecules. This extraction is commonly used for pesticides [13].

Thereafter, QuEChERS was tested and introduced as the sample preparation protocol for other LC-MS/MS methods as well [16,17,18]. Recently published papers and standards describe QuEChERS employed in the analysis of mycotoxins, ergot alkaloid or *Alternaria* toxin [19,20,21,22,23]. Although the extraction of *Alternaria* toxins from food is preferably done with a methanolic medium, QuEChERS can successfully be applied for their extraction in various food samples as well [22]. Thus, QuEChERS has become a general method for sample preparation before LC-MS/MS analyses [24].

Even though the extraction of compounds having different structures and hydrophobicity could be carried out with QuEChERS, the simultaneous HPLC analysis of the target compounds needs thorough optimization because some toxins (e.g., ergot alkaloids or tenuazonic acid (TEA)) require alkaline pH conditions in the mobile phase [19,22,25,26]. However, the HPLC separation of acidic (e.g., ochratoxin, *Alternaria* toxins) or basic compounds (e.g., alkaloids) generally requires an acidic pH condition to obtain appropriate peak shape and resolution [27]. Furthermore, MS detection in the positive ionization mode yields better sensitivity with an acidic mobile phase composition as the precursor ions are generally protonated molecules ([M+H]^+^). Consequently, multi-methods published earlier focused on compounds that could be separated with acidic or neutral eluents [10,11,12,13,28,29,30,31] and excluded those compounds that required alkaline conditions.

Six ergot alkaloids and their corresponding six epimers are currently regulated in the EU [1]. The simultaneous analysis of the twelve compounds requires an alkaline mobile phase pH to obtain baseline separation between the ergot alkaloids and their corresponding epimers [19,25]. TEA belongs to the *Alternaria* toxin group [3]; it is a chelating compound and forms complexes at acidic pH with metal ions occurring in the eluent [32,33]. Therefore, its LC-MS separation needs either pre-column derivatization or alkaline (pH > 8) conditions in the eluent [32,33,34,35,36] to include in a multi-method. Consequently, a multi-method that includes TEA and all regulated ergot alkaloids along with other mycotoxins and pesticides has not been published yet.

The aim of the present study was to develop for the first time a multi-method that allows for the analysis of food contaminants such as pesticides and toxins as well as alkaloids and *Alternaria* toxins including TEA. Therefore, the pH of the mobile phase has been optimized so that the HPLC separation allows fit-for-purpose chromatographic resolution for analyzing ergot alkaloids together with their epimers a functional peak shape for challenging TEA. In addition, appropriate sensitivity for MS/MS detection carried out with polarity switching had to be optimized based on the mobile phase condition. A further goal of the paper was to verify the method with validation at low concentration levels and to evaluate the accuracy of the method involving a number of QC and PT sample analyses. Finally, the results of the multi-method on QC samples were compared to those obtained after analysis of the samples with official standard methods.

## 2. Results and Discussion

### 2.1. LC-MS/MS Method

In the analysis of pesticides and mycotoxins, the HPLC separation is generally done under acidic pH or sometimes at neural pH conditions [12,37,38]. In contrast, the separation of alkaloids and the *Alternaria* toxins requires alkaline pH conditions in the eluent recommended by the standards [19,25,26]. Hence, the optimal pH condition must be obtained at a weak alkaline pH to achieve fit-for-purpose separation of all compounds in the developed method. The pH conditions between 8.0 and 8.8 were therefore tested since the TEA gives a distorted peak shape below pH 8.0 using an HPLC column packed with C18 material, and the pH limit of the HPLC column utilized was at pH 9.0.

The EU standard methods recommend pH 10.0 to analyze ergot alkaloids in order to obtain an appropriate peak resolution between ergot alkaloids and their corresponding epimers, otherwise, peak interference may occur due to isobaric ion transitions [19,25]. On the other hand, an alkaline pH can decrease the sensitivity of those compounds ionized in the protonated molecule form. With these limitations in mind, the pH of the aqueous mobile phase was increased stepwise (in 0.1 unit increments) from 8.0 to 8.8. At pH 8.8, some pesticides displayed low intensity, e.g., cypermethrin, cyprodinil, pendimethalin and permethrin. However, this pH produced better resolution for ergot alkaloids. The lowest limit of pH in which the baseline separation could be achieved between ergot alkaloids and their corresponding epimers was at 8.3 (Figure 1). This alkaline pH did not considerably influence the retention and sensitivity of mycotoxins. Only ochratoxin A (OTA) and the fumonisins (FB1, FB2 and FB3) had retention time shifts between pH 8.0 and 8.8. The sensitivity of the detection of mycotoxins, carried out in a positive ionization mode, did not decrease under alkaline pH conditions (Figure 2 and Figure 3) compared with the acidic conditions detailed in the standard method [12].

In our earlier studies, we found that the mobile phase did not require acidic conditions to obtain high sensitivity for the analysis of pesticides and mycotoxins using LC-MS/MS separation and employing positive ionization [37,39]. The response of DON, aflatoxins and some pesticides (e.g., chlorpyrifos-ethyl/methyl), detected as a protonated molecule ion, slightly increased at alkaline pH in comparison with acidic conditions (Figure 2). This may be caused by the sodium content of the utilized HPLC water. The higher sodium level in water yields sodium formate in the eluent when formic acid is used for acidification, and this salt can decrease the ionization of protonated molecules in the ion source. Again, the non-acidified eluent caused sensitivity drops for only a few compounds, but rather enhanced the sensitivity for most of the molecules, which resulted in fit-for-purpose sensitivity for all compounds pursued for analysis in the developed method.

The other aspect of the method was the sample preparation. The QuEChERS approach described for maize and wheat samples in the standard pesticide method was tested since the QuEChERS is also used for ergot alkaloids in the EN 17425:2021 standard method [19]. Moreover, Bessaire et al. published a collaborative trial using the QuEChERS-LC-MS/MS method for analyzing mycotoxins [23] and Mujahid et al. proposed QuEChERS for Alternaria toxins [22]. The modification to the standard pesticide method involved the extraction time, which was adjusted to 30 min. Even though QuEChERS has been tested for analysis of Alternaria toxins [22], the acetonitrile-based extraction was not recommended for the extraction of Alternaria toxins earlier because methanol is preferable [40]. We also found that the absolute recovery of polar TEA, using QuEChERS extraction, was lower than 50%. The other two Alternaria toxins, AME and AOH, had higher recoveries (>80%) due to their lipophilic structures. Hence, the Alternaria ISTD solution (AME-d3, AOH-d2, TEA-^13^C2) was added to the sample before extraction to enhance the recovery. This was also recommended in previous papers [22,40]. The isotope dilution considerably improved the recovery of TEA, but the 50% loss increased its limit of quantification (LOQ). Furthermore, the injection solution consisted of acetonitrile, which caused peak distortion of TEA when the injection volume was higher than 2.0 µL. The separation of Alternaria toxins at pH 8.3 in a cereal matrix gave fit-for-purpose LC-MS/MS analysis (Figure 4), but the high (>50%) ion suppression of AME, described also in earlier methods [34,35,36,40], was also seen. Therefore, isotope dilution was needed for appropriate quantification.

### 2.2. Method Evaluation

During the method evaluation, twenty-three QC and PT samples (see Appendix A) were analyzed along with validation samples (spiked blanks). The validation samples involved maize and wheat matrices (Table 1). These blank samples were spiked at two levels with 15 replicates (see Section 3.6) to evaluate the recovery and precision (Table 1). The method evaluation was done only for those compounds possessing corresponding isotopically labeled ISTD. In the case of pesticides (excluding chlorpyrifos-ethyl) and ergot alkaloids, their signals were not compensated by ISTD, so only screening and semi-quantitative analysis could be done. Even though the EN 15662:2018 and EN 17425:2021 standards allow for quantification of pesticides and ergot alkaloids using neat solvent calibration, our experience was that this leads to considerable overestimation of the pesticide concentration in spiked samples. When analyzing ergot alkaloids, low recovery was observed. The higher and lower recovery was caused by ion enhancement and ion suppression, respectively. However, the QC sample analysis gave acceptable results for both pesticides and ergot alkaloids due to their broader satisfactory range. In agreement with the SANTE 11312/2021 guideline [41], standard addition is the appropriate quantification approach for pesticides and this approach is also suggested by the EN 17256:2019 standard method for ergot and tropane alkaloids. Confirmatory analyses have been performed according to standard methods using the standard addition approach, and the results were satisfactory (Table 2). In this validation, the screening detection limit (SDL) was set for pesticides and ergot alkaloids [41]. The SDL was established as the lowest spiking level (10 µg/kg or 50 µg/kg) at which the signal-to-noise ratio (SNR > 10) and the ion ratios (within the 30% tolerance range) are acceptable. For all pesticides and ergot alkaloids, 10 µg/kg as SDL was appropriate.

The recovery calculated for those compounds listed in Table 1 was not lower than 67.1% (TEA at 500 µg/kg level) and generally ranged between 70.0% and 111%. According to the standard guideline for the determination of mycotoxins [42], recovery between 50% and 120% is acceptable with precision below 30%. These satisfactory ranges are also applicable for the analysis of alkaloid toxins and Alternaria toxins. The validation data met the standard. The validation results for analyzing chlorpyrifos-ethyl also met the SANTE requirements [41].

In Table 2, we summarized the results obtained after the application of the multi-method on several naturally contaminated or spiked QC and PT samples. The concentrations of contaminants in these samples were also evaluated using the individual EU standard methods. Based on the assigned/reference values and their target standard deviations (Appendix A), the Z-score for the concentrations evaluated with the multi-method was calculated. Generally, the Z-score is satisfactory between −2 and +2.

In total, there were 11 samples analyzed for mycotoxins, of which 4 were PT samples. The PT samples had maize and wheat matrices in which aflatoxins, DON, fumonisins, HT-2, T-2, OTA and ZON could be detected, so all mycotoxins involved in the method was found at least in one PT sample. The evaluations of aflatoxins were successful at both low (below µg/kg) and medium level (sub-µg/kg) concentrations. The quantification of other mycotoxins was also satisfactory. The results obtained with the multi-method were close to those obtained by the standard methods [12]. This was also true for the seven QC samples. The quantification using the multi-method, which utilizes an alkaline mobile phase separation, QuEChERS sample preparation and isotope dilution with ^13^C labeled analogs, resulted in satisfactory analysis for all mycotoxins in the naturally contaminated samples. In total, forty-six Z-scores were evaluated, and they were found to be between −1.67 and 1.96. Generally, the alkaline condition did not influence the analysis of mycotoxins. Retention shifts for fumonisins and OTA were observed, but the quantification was not affected by the different background. Isotope dilution with ^13^C labeled standards further improved the quantification of mycotoxins.

Two QC (wheat and kidney bean) and two wheat PT samples were analyzed for pesticides (Figure 5). The QC samples contained multi-residues while the PT sample was contaminated with only chlorpyrifos-ethyl. The quantification of chlorpyrifos-ethyl was successfully carried out by the multi-method. The standard method used isotope dilution. However, the QC sample analysis showed underestimation of flufenoxuron and isofenphos-methyl in kidney bean QC and a questionable concentration of dimethoate and pirimiphos-methyl in wheat QC. This is caused by the pure solvent calibration, which could not compensate for the background and the recovery. The standard addition approach used for the confirmatory analysis gave satisfactory data for analyzing pesticides. In its current form, the multi-method utilizing neat solvent calibration can only be used as a screening approach.

In the case of alkaloids, five QC samples were analyzed. Two rye samples were spiked with ergot alkaloids, while the three cereal samples were naturally contaminated with tropane alkaloids. The samples were evaluated with both the multi-method and the standard method (EN 17256:2019). The alkaline mobile phase condition (pH 8.3) allowed for the baseline separation of the ergot alkaloids and their corresponding epimers. Therefore, all 12 compounds could be separated in all samples. In the separation of tropane alkaloids (atropine and scopolamine), isotope dilution produced satisfactory results for all samples. However, the pure solvent calibration used in the multi-method resulted in two non-satisfactory data sets in the two QC samples for analyzing ergot alkaloids (ergosine/inine and ergotaminine). Again, the standard addition approach used in the standard method for the analysis of ergot alkaloids produced successful results (Table 2).

In the case of *Alternaria* toxins (AME, AOH and TEA), the accuracy must be improved by spiking the isotopically labeled ISTDs at the beginning of sample preparation. This is critically needed because of the use of the acetonitrile-based extraction in QuEChERS. The TEA had mostly low recovery compared to the standard method using methanolic extraction. Two naturally contaminated wheat and a sunflower seed sample were analyzed using the multi-method and the standard method (EN 17521:2021). The detected concentrations compensated by isotope dilution gave satisfactory concentrations with both methods in all samples (Table 2). However, the standard method may be preferable over the method presented here because it gives better LOQ due to the higher injection volume.

## 3. Materials and Methods

### 3.1. Reagents and Samples

Dried-down analytical standards such as *Alternaria* toxins (100 μg), ergot alkaloids (500 μg), tropane alkaloids (100 μg), mycotoxin stock solutions and ^13^C isotopically labeled stock solutions were obtained from Romer Labs (Tulln, Austria). Stock solutions were prepared by adding 1.0 mL methanol (*Alternaria* toxins), 1.0 mL acetonitrile (tropane alkaloids), 5.0 mL acetonitrile (ergot alkaloids) to the vial and standards were redissolved in the solvent to obtain a concentration of 100 µg/mL. Stock solutions were kept at –18 °C for a year. Deuterated isotopically labeled standards (*Alternaria* toxins: AME-d3, AOH-d2, TEA-^13^C2; tropane alkaloids: atropine-d5 and scopolamine-^13^C1-d3; pesticide: chlorpyrifos-d10), Pesticide Mixture 167 and Pyrethroide Pesticide Mixture 153 were acquired from LGC (Wesel, UK). Piperonyl butoxide (a synergistic component), permethrin and diphenylamine individual standards were purchased from the Merck-Sigma group (Schnelldorf, Germany). An LC-MS comprehensive pesticide mixture containing 253 compounds was purchased from Agilent Technologies (Waldbronn, Germany). Stock solutions (1 mg/mL) for individual standards were prepared and stored by following the procedure given in the pesticide database [43].

Methanol, acetonitrile, ammonium formate, formic acid, ammonia solution (25%), (either LC-MS or HPLC grade) and the Ascentis Express C18 HPLC column (100 mm × 3 mm, 2.7 µm) were purchased from the Merck-Sigma group (Schnelldorf, Germany). The EN 15662:2018 QuEChERS extraction salt (4 g MgSO4, 1 g NaCl, 1 g Na-citrate × 2H_2_O and 0.5 g) and HPLC pre-column holders and C18 pre-column cartridges (4 mm × 3 mm; 5 µm) were obtained from Phenomenex (Torrance, CA, USA). Honeywell HPLC-gradient-grade water was acquired from Thomasker (Debrecen, Hungary). The final aqueous mobile phase (solvent A, pH 8.3) was prepared by adding 65 µL ammonia solution to 1 L HPLC water containing 5 mM ammonium formate.

PT and QC samples (23 in total) were obtained from various companies and details are provided in Appendix A.

### 3.2. Instrumentation

LC-MS/MS analyses were carried out using an Agilent 6470B triple quad consisting of an Agilent 1260 liquid chromatograph coupled to a 6470B MS detector equipped with an Agilent JetStream ion source (Agilent Technologies (Waldbronn, Germany)). Data acquisition and evaluation were performed with the Masshunter software version 10.1.

### 3.3. Sample Preparation

Samples were ground (<1 mm) before extraction and thoroughly mixed to assure adequate homogeneity. The sample preparation was based on the standard QuECHERS approach for regular pesticide residue analysis in cereals [13], with mechanical shaking modification to obtain appropriate extraction for the toxins. Samples (5.0 g) were weighed in a 50 mL plastic centrifuge tube and 100 µL Alternaria ISTD solution (25 µg/mL TEA-^13^C2, 5.0 µg/mL AOH-d2 and AME-d3 in methanol) was spiked into the sample. Then, 10.0 mL distilled water was added to the samples, followed by 9.9 mL acetonitrile. The extraction was carried out with a laboratory shaker (CAT S50, CAT M. Zipperer GmbH, Ballrechten-Dottingen, Germany) at full speed (600 min^−1^) for 30 min. After the extraction, EN 15662:2018 QuEChERS salt mixture was added and the samples were hand-shaken for 1.0 min, followed by centrifugation at 2300× *g* (Thermo Megafuge 16, Unicam Kft, Budapest, Hungary) at ambient temperature. Then, 470 µL of the upper layer and 30 µL of ISTD solution (AFB1-^13^C17, AFB2-^13^C17, AFG1-^13^C17, AFG2-^13^C17, OTA-^13^C20 in 15 ng/mL; T-2-^13^C24, HT-2-^13^C22, ZON-^13^C18 in 150 ng/mL; DON-^13^C15, FB1-^13^C34, FB2-^13^C34, FB3-^13^C34, chlorpyrifos-ethyl-d10 in 300 ng/mL; atropine-d5, scopolamine-^13^C1-d3 in 100 ng/mL in 50% acetonitrile) were mixed in a 2.0 mL screw-cap HPLC vial and vortexed prior to injection into the LC-MS/MS instrument.

### 3.4. LC-MS/MS Separation

Compounds were separated on an Ascentis Express C18 HPLC column (100 × 3 mm, 2.7 µm) equipped with a C18 guard column (4 mm × 3 mm, 5 µm) Merck-Sigma group (Schnelldorf, Germany). The binary gradient elution mode was applied with solvent A containing 5 mM ammonium formate in water (pH 8.3) and solvent B containing methanol. The mobile phase gradient consisted of 5% B at 0 min; 5% B at 0.5 min; 40% B at 3.0 min; 100% B at 15 min; 100% B at 19 min; 5% B at 19.1 min; 5% B at 26.0 min; flow rate was set to 0.5 mL/min. The column thermostat and autosampler were maintained at 39 °C and at 18 °C, respectively. The injection volume was 2.0 µL. Compounds were detected using positive/negative ionization mode and dynamic multiple reaction monitoring (dMRM) scan mode. Ion transitions for 295 compounds are presented in Appendix A. The MRM time window was 60 s, and the cycle time was 1000 ms. The Agilent Jet Stream ion source parameters were as follows: drying gas temperature, 300 °C; sheath gas temperature, 350 °C; nebulizer, 35 psi; gas flow, 7 L/min; sheath gas flow, 11 L/min; capillary voltage, ± 3500 V; and nozzle voltage, +0, −1000 V. The HPLC effluent was directed into waste from 0 to 2.0 min.

### 3.5. Quantification

Calibrants in 50% acetonitrile were prepared from the native working standard mixture along with ISTD solutions, considering the dilution factor (2.13*×*) of the sample preparation. The calibration levels, expressed in µg/kg, are detailed in Table 3. For some pesticides, the lowest calibration level was 0.2 µg/kg, however, for most of them, 1 µg/kg could be used as the starting point of the calibration. Only those compounds that possessed the corresponding isotopically labeled analogs could be appropriately quantified. These were the mycotoxins, the tropane alkaloids, the Alternaria toxins, and chlorpyrifos-ethyl. Even though the pesticide (EN 15662:2018) and ergot alkaloid (EN 17425:2021) standard methods allow the quantification with neat calibrants [13,19], the presented method works only as a screening approach for them in the absence of ISTD. In the case where compounds from their group are identified, a further quantification using the standard addition approach is needed, in accordance with the SANTE 11312/2021 guidelines and EN 17256:2019 standard method [25,41].

The concentrations of analytes could be directly obtained from the equations of linear calibration weighted with the factor of 1/x. The determination coefficients obtained under the validation study were not lower than 0.9950.

### 3.6. Validation

The confirmatory validation was performed for the mycotoxins, the tropane alkaloids, the *Alternaria* toxins and chlorpyrifos-ethyl. The recovery and precision were calculated from the analysis of spiked maize and wheat samples. The fortified samples were prepared on three different days at two concentration levels (level 1 and level 2) by the operators (Table 1). In total, fifteen samples were analyzed. The levels were: AFB2, AFG2—0.25 µg/kg and 1.25 µg/kg; AFB1, AFG1, atropine and scopolamine—1.0 µg/kg and 5.0 µg/kg; AME and AOH—2 µg/kg and 10 µg/kg; HT-2, T-2 and ZON—5 µg/kg and 25 µg/kg; chlorpyrifos-ethyl, DON, FB1, FB2 and FB3—10 µg/kg and 50 µg/kg; TEA—200 µg/kg and 1000 µg/kg.

Ergot alkaloids and pesticides were also spiked into the samples along with the other compounds mentioned above. Their levels were 10 µg/kg and 50 µg/kg; however, appropriate quantification could not be performed due to the absence of background compensation with ISTD. Hence, the validation of these samples was performed as a screening validation, and the SDL (either 10 µg/kg or 50 µg/kg) was evaluated. The LOQ was set as the lowest calibration point.

## 4. Conclusions

A novel LC-MS/MS multi-method has been developed for analyzing toxins and pesticides together. The sample preparation is the modification of the QuEChERS-based approaches described in the pesticide and ergot alkaloid standard method or developed by Mujahid et al. (2020) or Bessaire et al. (2019). The chromatographically challenging TEA could be included in the method along with all regulated ergot alkaloids by using alkaline mobile phase conditions. The pH of the eluent did not influence the analysis of pesticides and mycotoxins. The method was evaluated by analyzing several QC and PT samples. Moreover, results obtained with the multi-method was compared with those data obtained by the individual EU standard methods. Even though the results with the standard methods are better, similarly good data can be obtained with the multi-method, which covers 295 compounds and unites five standard methods.

## Figures and Tables

**Figure 1 molecules-28-01468-f001:**
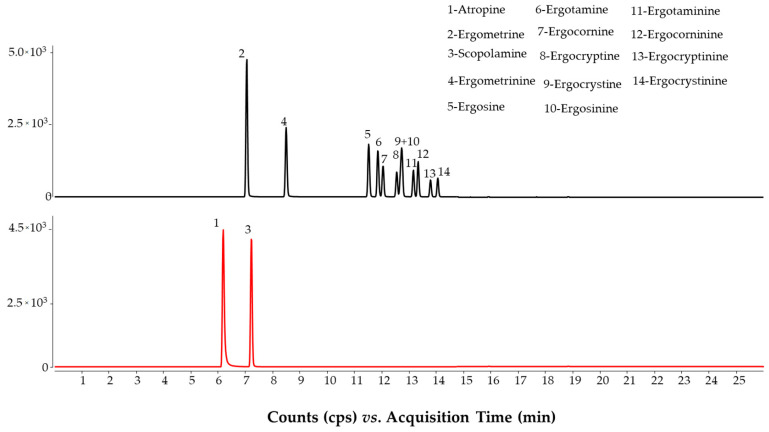
Separation of ergot (10 µg/kg) and tropane (5 µg/kg) alkaloids in spiked wheat samples using the optimized method.

**Figure 2 molecules-28-01468-f002:**
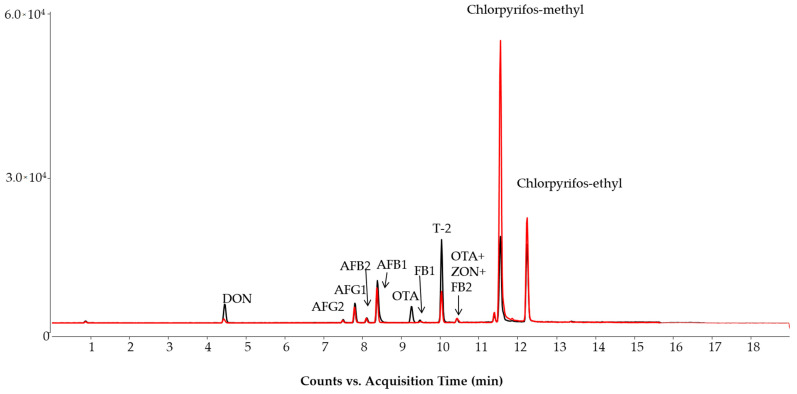
Total ion chromatograms of mycotoxins and chlorpyrifos-ethyl/methyl in standard solution recorded under acidic (pH 3, red line) and alkaline (pH 8.3, black line) mobile phase conditions. Concentrations: chlorpyrifos-ethyl/methyl, DON, FB1, FB2, 10 ng/mL; AFB1, AFG1, OTA, 1 ng/mL; AFB2, AFG2, 0.25 ng/mL; HT-2, T-2, ZON, 5 ng/mL. The chromatograms were recorded in an earlier stage of the method development.

**Figure 3 molecules-28-01468-f003:**
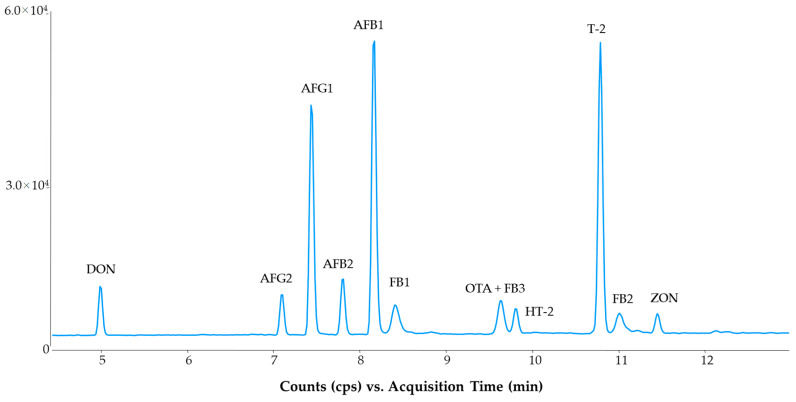
Separation of mycotoxins in wheat samples using the optimized method. Concentrations: DON, FB1, FB2, FB3, 50 µg/kg; AFB1, AFG1, OTA, 5 µg/kg; AFB2, AFG2, 1.25 µg/kg; HT-2, T-2, ZON, 25 µg/kg.

**Figure 4 molecules-28-01468-f004:**
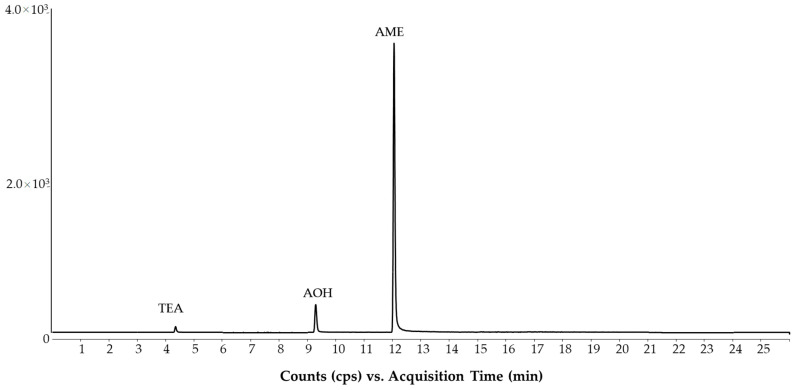
Separation of *Alternaria* toxins in wheat samples using the optimized method. Concentrations: AOH and AME, 2 µg/kg; TEA 100 µg/kg.

**Figure 5 molecules-28-01468-f005:**
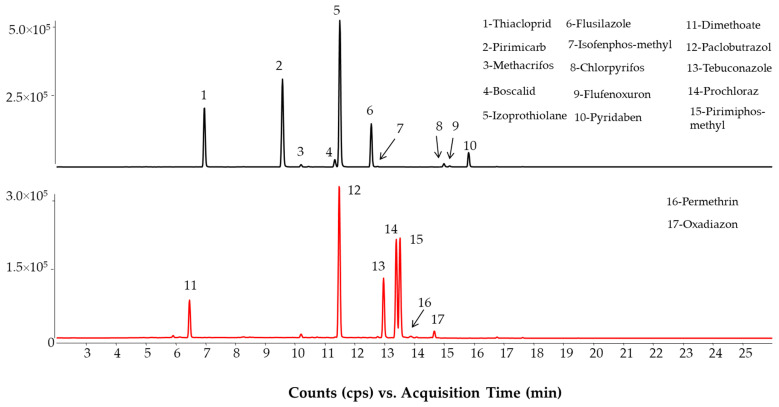
Separation of pesticides in kidney bean (T09133QC, above) and wheat flour (T09140QC, below) FAPAS QC samples using the multi-method. Concentrations are detailed in Table 2.

**Table 1 molecules-28-01468-t001:** Validation results for maize and wheat samples. Spiking levels are summarized in Section 3.6.

Components	Repeatability RSD% (*n* = 5)		Reproducibility RSD% (*n* = 15)		Recovery% (*n* = 15)		LOQ(µg/kg)	Linearity	
	Level 1	Level 2	Level 1	Level 2	Level 1	Level 2		Equation	R^2^
AFB1	4.79	4.21	8.30	6.24	88.9	85.8	0.20	0.8850x − 0.0985	0.9992
AFB2	11.1	11.0	23.8	19.9	111	89.3	0.05	0.6217x − 0.0876	0.9999
AFG1	11.9	4.48	13.7	16.7	88.7	94.3	0.20	0.6388x + 0.0125	0.9998
AFG2	19.1	13.6	26.5	10.4	119	101	0.05	0.4611x − 0.00985	0.9988
AME	5.31	6.05	5.32	4.82	87.4	86.3	0.20	0.0745x − 0.0253	0.9992
AOH	21.3	14.6	23.0	14.6	108	103	0.20	0.1455x − 0.0325	0.9999
Atropine	4.02	2.18	4.11	2.18	78.6	83.9	0.20	0.6052x + 0.0897	0.9992
Chlorpyrifos-ethyl	11.3	3.86	11.3	15.6	97.1	72.7	0.20	0.0407x + 0.0014	0.9994
DON	21.1	6.71	20.5	8.86	86.0	82.9	10.0	0.04475x − 0.0014	0.9995
FB1	10.1	5.7	12.3	9.7	83.1	73.9	10.0	0.1118x − 0.00547	0.9979
FB2	9.9	5.2	17.2	9.8	91.4	81.5	10.0	0.0954x − 0.0145	0.9999
FB3	7.7	6.4	13.0	10.1	95.3	90.5	10.0	0.0954x − 0.0145	0.9999
HT-2	17.7	7.89	26.8	12.8	102.6	92	5.0	0.0051x + 0.0011	0.9998
OTA	21.7	4.25	21.7	10.5	101	70.6	1.00	0.1045x + 0.0745	0.9983
Scopolamine	3.49	2.80	5.57	2.80	74.9	76.6	0.20	0.3750x−0.0455	0.9975
T-2	8.86	3.44	11.9	13.6	96.1	89.2	1.00	0.04459x + 0.0084	0.9994
TEA	12.1	11.8	24.2	28.2	100	67.1	200	0.0153x + 0.00632	0.9988
ZON	13.8	9.87	14.6	13.3	95.4	88.2	1.00	0.0397x − 0.00754	0.9998

AFB1: aflatoxin B1; AFB2: aflatoxin B2; AFG1: aflatoxin G1; AFG2: aflatoxin G2; AME: alternariol monomethyl ether; AOH: alternariol; DON: deoxynivalenol; FB1: fumonisin B1; FB2: fumonisin B2; FB3: fumonisin B3; OTA: ochratoin A; TEA: tenuazonic acid; ZON: zearalenone.

**Table 2 molecules-28-01468-t002:** The results of QC and PT sample analysis using the multi-method and EU standard methods.

Sample Code	Matrix	Detected Compounds	Detected Concentrations (µg/kg)	Calculated Z-Score	Evaluation	Detected Concentrations with Standard Method (Reference Value, µg/kg)	Evaluation
GAFTA PT 2022-M2	Maize	AFB1 AFB2 AFG1 AFG2 Total Aflatoxins	1.77 0.507 2.05 0.69 5.05	0.49 0.0 1.67 1.90 1.61	Satisfactory	1.79 (1.60) 0.524 (0.50) 1.86 (1.50) 0.552 (0.50) 4.73 (3.73)	Satisfactory
GAFTA PT 2022-M1	Wheat	HT-2 T-2	7.8 30.1	0.31 −1.42	Satisfactory	9.2 (7.3) 38.1 (43.8)	Satisfactory
Romer PT M22411 AF	Maize	AFB1 AFB2 AFG1 AFG2 Total Aflatoxins FB1 FB2 FB3 Total Fumonisins	8.35 0.576 0.706 − 9.73 1136 296 121 1553	−0.22 −0.38 1.96 − 0.0 −1.56 −1.21 −1.31 −1.55	Satisfactory	9.44 (8.79) 0.450 (0.63) 0.554 (0.49) − 10.4 (9.72) 1414 (1425) 402 (387) 168 (168) 1984 (1911)	Satisfactory
Romer PT M22161 DZO	Wheat	DON OTA ZON	2032 30.3 702	0.78 0.79 1.81	Satisfactory	1694 (1826) 27.5 (25.9) 519 (545)	Satisfactory
Romer QC M21161DZO	Wheat	DON OTA ZON	2597 25.0 200	−0.96 −0.84 0.62	Satisfactory	2802 (2841) 28.7 (30.7) 195 (177)	Satisfactory
EURL QC 2016 O161	Oat	HT-2 T-2	161 63.8	0.33 −0.41	Satisfactory	98 (150) 58.8 (70.3)	Satisfactory
EURL QC 2017 A004	Wheat	DON	434	−0.97	Satisfactory	388 (551)	Satisfactory
EURL QC 2016 C257	Maize	AFB1 DON FB1 FB2 ZON	10.9 553 501 237 210	0.13 −0.43 −1.58 0.27 1.33	Satisfactory	9.10 (10.6) 454 (618) 653 (768) 246 (224) 147 (162)	Satisfactory
Romer QC DZO10006460	Wheat	DON OTA ZON	618 7.4 34.6	−1.67 −1.24 −0.05	Satisfactory	859 (825) 7.2 (10) 34.4 (34.9)	Satisfactory
Romer QC 10003613	Maize	AFB1 AFB2	8.5 1.93	−0.59 −0.49	Satisfactory	8.6 (9.5) 2.5 (2.1)	Satisfactory
Trilogy QC TQC-MMF11-100	Maize	AFB1 AFB2 Total Aflatoxins DON FB1 FB2 FB3 Total Fumonisins HT-2 T-2 OTA ZON	20.8 1.32 22.1 1932 1168 442 95 1705 121.9 104.5 17.5 374	0.54 0.07 0.50 0.16 −1.55 1.2 −0.2 −0.68 −0.41 1.22 −0.31 0.35	Satisfactory	17.6 (18.6) 1.20 (1.30) 18.8 (19.9) 1758 (1900) 1276 (1400) 366 (400) 93 (100) 1735 (1900) 149 (127) 92.5 (94.8) 22.6 (18.5) 373 (360)	Satisfactory
FAPAS QC T09133QC	Kidney Beans (Dried)	Boscalid Chlorpyrifos Flufenoxuron Flusilazole Isofenphos-methyl Isoprothiolane Methacrifos Pirimicarb Pyridaben Thiacloprid	81 143 26 119 25 154 70 96.4 32.3 67.2	−1.12 1.74 −2.38 −1.09 −3.06 −0.06 −1.88 −0.21 −1.79 −0.95	Questionable	94 (107) 68 (103) 38.5 (55) 152 (155) 80 (77) 142 (156) 118 (119) 82 (101) 69 (53) 62 (85)	Satisfactory
FAPAS QC T09140QC	Wheat flour	Dimethoate Oxadiazon Paclobutrazol Permethrin Pirimiphos-methyl Prochloraz Tebuconazole	47.1 101 134 28.1 104 166 79	2.10 1.35 1.02 −2.57 0.00 0.40 0.44	Questionable	41.1 (32.1) 65.4 (77.8) 112 (98.3) 66.8 (64.7) 99.3 (104) 111 (153) 101 (87.3)	Satisfactory
PT, Chlorpyrifos-ethyl	Wheat	Chlorpyrifos-ethyl	28.0	−0.68	Satisfactory	27.0 (33.0)	Satisfactory
PT, Chlorpyrifos-ethyl	Wheat	Chlorpyrifos-ethyl	17.0	0.0	Satisfactory	19.6 (16.9)	Satisfactory
EURL 2017 QC EA047	Rye	Ergocornine/inine α-Ergocryptine/inine Ergocrystine/inine Ergometrine/inine Ergosine/inine Ergotamine/inine	294 304 676 92.1 136 641	0.00 +1.60 −0.55 −1.20 −2.65 −0.45	Questionable	280 (295) 337 (231) 651 (752) 114 (116) 222 (242) 606 (695)	Satisfactory
FAPAS QC 22180	Rye	Ergocornine Ergocorninine α-Ergocryptinine Ergocrystine Ergocrystinine Ergometrine Ergometrinine Ergosine Ergotamine Ergotaminine Total Ergot Alkaloides	45.2 12.6 13.9 85.6 20.8 27.1 4.30 16.9 34.5 6.03 338	1.79 0.07 −1.30 −0.91 −1.95 0.21 0.11 −1.08 −1.42 −2.29 −1.06	Questionable	40.7 (32.4) 15.8 (12.4) 15.7 (19.5) 141 (107) 23.6 (36.4) 32 (25.9) 4.64 (4.2) 19.1 (22.2) 41.6 (50.2) 18.0 (13.6) 353 (419)	Satisfactory
EURL QC 2016 C029	Cereal	Atropine Scopolamine	0.81 0.111	−1.37 −1.85	Questionable	1.11 (1.16) 0.169 (0.183)	Satisfactory
FAPAS QC 22179	Cereal	Atropine Scopolamine	6.5 3.6	−1.53 −0.37	Satisfactory	8.82 (9.8) 4.83 (3.88)	Satisfactory
EURL QC 2016 E087	Cereal	Atropine Scopolamine	6.7 0.63	−0.45 −1.77	Satisfactory	6.29 (7.44) 0.76 (1.03)	Satisfactory
QC 2018T15	Wheat	AME AOH TEA	4.0 3.3 76.0	−0.95 −1.63 2.1	Satisfactory	4.08 (5.06) 5.96 (5.11) 71.0 (52.0)	Satisfactory
QC 2018 B56	Wheat	AME AOH TEA	0.78 1.51 180	−1.53 −1.28 −1.95	Satisfactory	0.69 (1.17) 2.51 (2.1) 314 (297)	Satisfactory
QC 2018 X06	Sunflower seed	AME AOH TEA	1.68 1.57 87	−0.91 −1.06 −1.84	Satisfactory	2.01 (2.1) 1.32 (2.06) 102 (146)	Satisfactory

The outlier results are highlighted with red.

**Table 3 molecules-28-01468-t003:** Calibration levels.

Compounds	Cal 1 (µg/kg)	Cal 2 (µg/kg)	Cal 3 (µg/kg)	Cal 4 (µg/kg)	Cal 5 (µg/kg)	Cal 6 (µg/kg)
AFB1	0.2	1	2	10	20	50
AFB2	0.05	0.25	0.5	2.5	5	12.5
AFG1	0.2	1	2	10	20	50
AFG2	0.05	0.25	0.5	2.5	5	12.5
AME	0.2	1	2	10	20	50
AOH	0.2	1	2	10	20	50
Atropine/scopolamine	0.2	1	2	10	20	50
DON	10	50	100	500	1000	2500
Ergot alkaloids	0.2	1	2	10	20	50
Fumonisins	10	50	100	500	1000	2500
HT-2/T-2	1	5	10	50	100	250
OTA	1	5	10	50	100	250
Pesticides	0.2	1	2	10	20	50
TEA	100	500	1000	5000	10,000	25,000
ZON	1	5	10	50	100	250

## Data Availability

Not applicable.

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
