# Peer review of "Development of a Novel LC-MS/MS Multi-Method for the Determination of Regulated and Emerging Food Contaminants Including Tenuazonic Acid, a Chromatographically Challenging Alternaria Toxin"

_molecules, 2023, doi:10.3390/molecules28031468_

Round 1
Reviewer 1 Report
The authors have designed a single method for analysis of food contaminants using LC-MS/MS. This is of very high importance given the regulations surrounding the contaminants the their impacts on humans. The paper is of high importance and the following comments should be addressed by the authors in order to further improve the quality of the paper.
1. Page 1, Line 43: This sentences merely restates what is told in the previous sentences, so please remove this sentence.
2. Page 2, Line 88: The authors mention that acidic or basic analytes typically require acidic pH for good peak shape and resolution. Can the authors provide any reference for this? The mobile phase pH actually determines the charge state which thereby determines the interaction with the stationary phase. Altering the mobile phase pH can lead to different retention of both acidic and basic analytes.
3. Page 3, Line 137 and Figure 2: Authors mention that there was no decrease in sensitivity under alkaline pH conditions. Can the authors add the comparative figure so that this can be established.
4. Page 11, Line 327: The authors have mentioned the dimensions of the guard column only. Please add dimensions of the actually C18 column used.
5. Page 11, Line 333: How did the authors arrive at the optimized temperature of 39℃. Can the authors provide a figure of how the separation is affected at a lower or higher temperature?
6. Page 11, Line 342: According to my knowledge neat solvent means pure single solvent. The authors have mentioned in parenthesis that it is 50% acetonitrile. Can they clarify this statement?
Author Response
The authors have designed a single method for analysis of food contaminants using LC-MS/MS. This is of very high importance given the regulations surrounding the contaminants the their impacts on humans. The paper is of high importance and the following comments should be addressed by the authors in order to further improve the quality of the paper.
Answer: Thank you, we addressed all comments in the revised paper.
- Page 1, Line 43: This sentences merely restates what is told in the previous sentences, so please remove this sentence.
Answer: Thank you, we deleted this sentence in the revised paper.
- Page 2, Line 88: The authors mention that acidic or basic analytes typically require acidic pH for good peak shape and resolution. Can the authors provide any reference for this? The mobile phase pH actually determines the charge state which thereby determines the interaction with the stationary phase. Altering the mobile phase pH can lead to different retention of both acidic and basic analytes.
Answer: Thank you, yes, it is right. We added a reference here as the reviewer suggested.
- Tölgyesi, Á.; Berky, R.; Békési, K.; Fekete, S.; Fekete, J.; Sharma, V.K. Analysis of sulfonamide residue in real honey samples using liquid chromatography with fluorescence and tandem mass spectrometry detection. J. Liq. Chromatogr Relat. Technol. 2013, 36, 1105-1125. https://doi.org/10.1080/10826076.2012.685911
- Page 3, Line 137 and Figure 2: Authors mention that there was no decrease in sensitivity under alkaline pH conditions. Can the authors add the comparative figure so that this can be established.
Answer: Thank you, we added a comparative figure here.
- Page 11, Line 327: The authors have mentioned the dimensions of the guard column only. Please add dimensions of the actually C18 column used.
Answer: Thank you, we added the dimensions of the column here.
- Page 11, Line 333: How did the authors arrive at the optimized temperature of 39℃. Can the authors provide a figure of how the separation is affected at a lower or higher temperature?
Answer: Thank you, there was no optimization on column temperature, we used the temperature that is proposed in the standard methods. We did not record chromatograms at lower or higher temperatures. Generally, it can be stated that the lower temperature could increase the retention time together with the analysis time, but improves the peak resolution. Higher temperature can shorten the run time, but also decreases the peak resolution.
- Page 11, Line 342: According to my knowledge neat solvent means pure single solvent. The authors have mentioned in parenthesis that it is 50% acetonitrile. Can they clarify this statement?
Answer: Thank you, we deleted the „pure solvent” here.
Reviewer 2 Report
Research has an important value for developing a novel multi method for analyzing toxins and pesticides.
The introduction provides clear insight for what they expected from the study
purpose of the study was clear.
This research has an important value for developing a novel LC-MS/MS multi methods for analyzed the toxin and pesticides .
I think the conclusion of this study align with aims and procedures. and the cited references are relevant to the research
I see the paper reads well, also the percentage of scientific citation is appropriate, as you can see in the attachment file
Good luck for the researcher

Author Response
Research has an important value for developing a novel multi method for analyzing toxins and pesticides.
The introduction provides clear insight for what they expected from the study
purpose of the study was clear.
Answer: Thank you.
Reviewer 3 Report
This manuscript provides a novel LC-MS/MS multi-method to analyze toxins and pes-382 ticides together. The results are Compellent, I recommend publishing.
Author Response
This manuscript provides a novel LC-MS/MS multi-method to analyze toxins and pes-382 ticides together. The results are Compellent, I recommend publishing
Answer: Thank you.